# Phase diagram of the $\mathbb{Z}_3$-Fock parafermion chain with pair hopping

**Iman Mahyaeh[1⋆], Jurriaan Wouters[2†] and Dirk Schuricht[2‡]**

**1** Department of Physics, Stockholm University, SE-106 91 Stockholm, Sweden
**2** Institute for Theoretical Physics, Center for Extreme Matter and Emergent Phenomena,
Utrecht University, Princetonplein 5, 3584 CE Utrecht, The Netherlands

⋆ iman.mahyaeh@fysik.su.se, ⋆ j.j.wouters@uu.nl, ⋆ d.schuricht@uu.nl

## Abstract

We study a tight binding model of $\mathbb{Z}_3$-Fock parafermions with single-particle and pair-hopping terms. The phase diagram has four different phases: a gapped phase, a gapless phase with central charge $c=2$, and two gapless phases with central charge $c=1$. We characterise each phase by analysing the energy gap, entanglement entropy and different correlation functions. The numerical simulations are complemented by analytical arguments.

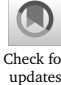
# 1 Introduction

Particles in three dimensions are known to be either bosons or fermions, distinguished by the symmetry or antisymmetry of their wave functions $\Psi(x_1, x_2)$ under particle exchange, ie, $\Psi(x_1, x_2) = \pm\Psi(x_2, x_1)$. This distinction has drastic consequences: Bosons can macroscopically occupy a quantum state, thus forming a Bose–Einstein condensate. In contrast, the asymmetry of the fermionic wave function implies the Pauli principle, which for example underlies the fundamentally different properties of metals and insulators.

However, the situation is completely different in lower dimensional systems, where more complicated (and thus more interesting) quantum statistical properties become possible. Since low-dimensional systems are ubiquitous in condensed-matter physics—think of two-dimensional systems like graphene [1] or two-dimensional electron gases in quantum Hall transistors [2], one-dimensional quantum wires [3], or the dimensional restriction of ultracold atomic gases in optical lattices [4,5]—non-trivial quantum statistics has to be considered in these contexts.

One generalisation[1] of bosonic and fermionic statistics is provided by relaxing the symmetry requirement of the wave functions under particle exchange. Instead of symmetry or antisymmetry one allows wave functions satisfying $\Psi(x_1, x_2) = e^{i\theta}\Psi(x_2, x_1)$ with real angle $\theta$ [9]. In three spatial dimensions the double exchange of two particles is indistinguishable from the absence of exchanging the particles, thus implying $\theta = 0$ or $\theta = \pi$ as the only consistent choices. In two dimensions, however, any real value of the exchange angle $\theta$ is allowed, leading to the so-called (Abelian) anyon statistics [10,11]. Such anyons exist in the fractional quantum Hall effect: The collective excitations of this system have unusual properties like fractional charge [12] $e^* = e/3$ and anyonic statistics [13,14] with $\theta = \pi/3$, both properties have been observed in experiments [15,16].

Another generalisation of quantum statistics can be implemented starting from directly the Pauli principle [17,18]. The idea is to ask how the number of available quantum states $D$ will change if $\Delta N$ particles are added to the system,[2] with the statistical parameter $\alpha$ being defined as $\Delta D = -\alpha\Delta N$. In principle this concept can be defined in any spatial dimension, with bosons ($\alpha = 0$) and fermions ($\alpha = 1$) as special cases. Quasiparticles satisfying such a generalised exclusion statistics are for example spinon excitations in spin-1/2 chains [19–21]. We note that in a slightly simplistic way one can imagine particles satisfying generalised exclusion statistics with exclusion parameter $\alpha$ as being able to occupy a single quantum state with $1/\alpha$ particles.

There is a third generalisation, usually referred to as parafermions.[3] Historically these parafermions were introduced [22] to analyse clock models. The simplest quantum clock model can be obtained from an anisotropic limit of the two-dimensional classical three-state Potts model, which is a direct generalisation of the Ising model by allowing the degrees of freedom at the lattice sites to take one of three, or more generally $p$, different values. The resulting Hamiltonian of the quantum Potts chain is given by [23–26]

$$H_{\text{Potts}} = -J\sum_j(\sigma_j^\dagger\sigma_{j+1} + \sigma_{j+1}^\dagger\sigma_j) - f\sum_j(\tau_j^\dagger + \tau_j), \qquad J, f \geq 0. \tag{1}$$

Here the operators $\sigma_j$ and $\tau_j$ act on the three states of the local Hilbert space at lattice site $j$ and satisfy the algebra

$$\sigma_j^p = \tau_j^p = 1, \quad \sigma_j^\dagger = \sigma_j^{p-1}, \quad \tau_j^\dagger = \tau_j^{p-1}, \quad \sigma_j\tau_j = e^{2\pi i/p}\tau_j\sigma_j, \quad \sigma_i\tau_j = \tau_j\sigma_i \quad \text{for } i \neq j, \tag{2}$$

---

[1]Historically there were other attempts to generalise quantum statistics like Gentile's intermediate statistics [6] or Green's parafields [7,8].

[2]For simplicity we restrict ourselves to one particle species.

[3]Not to be confused with Green's parafield construction.

with $p = 3$. Similar to the transverse-field Ising chain, the quantum Potts chain (1) possesses a phase transition between a ferromagnetic phase for $f < J$ with three-fold degenerate ground state and a paramagnetic phase for $f > J$ with unique ground state. The two phases are separated by a quantum critical point which is described by the non-trivial conformal field theory [27, 28] with central charge $c = 4/5$.

Motivated by the recent interest in topological order and edge zero modes the quantum Potts chain (1) has received renewed interest. For $p = 2$ the Potts chain simplifies to the transverse-field Ising model and can thus be directly linked to the Kitaev chain [29], which constitutes the prototypical example for the appearance of Majorana edge zero modes. The Potts chain provides a natural generalisation thereof to interacting systems possessing parafermion edge modes [25]. Specifically, two parafermion operators $\gamma_{2j-1}$ and $\gamma_{2j}$ at each lattice site $j$ can be introduced via

$$\gamma_{2j-1} = \left( \prod_{k<j} \tau_k \right) \sigma_j, \qquad \gamma_{2j} = \omega^{(p-1)/2} \gamma_{2j-1} \tau_j, \qquad (3)$$

where $\omega = \exp(2\pi i/p)$ and the clock operators $\sigma_j$ and $\tau_j$ satisfy the algebra (2). For the parafermion operators this implies the relations

$$\gamma_j^p = \mathbf{1}, \qquad \gamma_j^\dagger = \gamma_j^{p-1}, \qquad \gamma_j \gamma_k = \omega^{\mathrm{sgn}(k-j)} \gamma_k \gamma_j, \qquad (4)$$

which for $p = 2$ simplify to the ones for Majorana operators, in particular the reality condition $\gamma_j^\dagger = \gamma_j$.

While parafermions have proven useful in statistical mechanics and the study of edge zero modes, they possess a huge drawback. Due to the relations (4) it is not possible to interpret $\gamma_j^\dagger$ as a particle creation operator at site $j$. Very recently this limitation was overcome by Cobanera and Ortiz [30] who introduced the so-called Fock parafermions (FPFs). Here the term "Fock" refers to the fact that the newly introduced operators $F_j^\dagger$ and $F_j$ can be interpreted as creation and annihilation operators for particles, which act on a Fock space in the sense that a definite number of particles at lattice site $j$ can be defined (see next section for the detailed definition). Hence FPFs constitute particles with anyonic and fractional exclusion statistics and thus provide the ideal framework to study the consequences of generalised quantum statistics on the properties of many-particle systems. In this work we will specifically investigate which types of many-particle states of FPFs exist in one-dimensional systems.

A first step in this direction has been taken very recently by Rossini et al. [31], who studied a tight-binding chain of FPFs simply hopping between neighbouring sites. For $p = 3$ (the case we will restrict ourselves) they uncovered a gapped phase reminiscent of a Mott insulator at unit filling, while at all other fillings a gapless anyonic Luttinger phase [32] emerged. In our work we will extend these results by generalising the simple hopping model to include also coherent hopping of two-particle pairs, which is possible as two FPFs may exist at the same lattice site. As a consequence of the pair hopping two additional phases appear in the phase diagram (see Figure 1): A second Luttinger phase (labeled R) and, between the two Luttinger phases, a gapless phase with central charge $c = 2$ (labeled M).

The paper is organised as follows. In the next section we review the construction of FPFs. In Section 3 we define the model and present its phase diagram, the main result of our paper. In Section 4 we explain the implementation of the numerical simulations, while in Section 5 we present our detailed results and analysis of the phase diagram. We conclude with a discussion in Section 6.

## 2 Fock parafermions

In this section we discuss Fock parafermions (FPFs) as introduced by Cobanera and Ortiz [30]. They appear as particle-like excitations constructed from parafermions in the same way as spinless fermions are obtained from Majorana fermions. To be more specific, let us start with discussing the concept of parafermions [22, 25], which can be viewed as a generalisation of Majorana fermions.

Consider a set of $2L$ parafermion operators $\gamma_j$ satisfying

$$\gamma_j \gamma_k = \omega^{\text{sgn}(k-j)} \gamma_k \gamma_j, \qquad \omega = \exp\left(\frac{2\pi i}{p}\right), \tag{5}$$

with integer $p \geq 2$. For $p = 2$ we obtain the simple anti-commutation relations of Majorana fermions, but for $p > 2$ the parafermions are neither commuting nor anti-commuting. The other relations fixing the algebra are

$$\gamma_j^{p-1} = \gamma_j^\dagger, \qquad \gamma_j^p = \mathbf{1}, \tag{6}$$

in which $\mathbf{1}$ is the identity operator. An explicit realisation is provided by (3).

As for Majoranas, for parafermions there is no notion of filling, ie, there are no highest and lowest weight states as we see from Equation (6). However, for Majorana fermions this can be remedied by introducing spinless Dirac fermions via

$$c_j = \frac{1}{2}(\gamma_{2j-1} + i\gamma_{2j}), \qquad c_j^\dagger = \frac{1}{2}(\gamma_{2j-1} - i\gamma_{2j}), \tag{7}$$

which then allow a direct interpretation as particle annihilation and creation operators.

In Reference [30] a similar particle description was introduced for parafermions. These so-called FPF operators are defined as

$$F_j = \frac{p-1}{p}\gamma_{2j-1} - \frac{1}{p}\sum_{m=1}^{p-1} \omega^{m(m+p)/2} \gamma_{2j-1}^{m+1} \gamma_{2j}^{\dagger m}. \tag{8}$$

They possess anyonic commutation relations on different sites,

$$F_j F_k = \omega^{\text{sgn}(k-j)} F_k F_j, \qquad F_j^\dagger F_k = \omega^{-\text{sgn}(k-j)} F_k F_j^\dagger, \qquad j \neq k, \tag{9}$$

which implies that their statistical angle is given by $\theta = 2\pi/p$, while on-site they satisfy

$$F_j^p = 0, \qquad F_j^{\dagger m} F_j^m + F_j^{p-m} F_j^{\dagger(p-m)} = \mathbf{1}, \quad m = 1, \ldots, p-1. \tag{10}$$

The Fock space can be constructed by acting with the creation operators on the vacuum state,

$$|n_1, n_2, \ldots, n_L\rangle = F_1^{\dagger n_1} F_2^{\dagger n_2} \ldots F_L^{\dagger n_L} |0\rangle. \tag{11}$$

Note that due to the first relation in (10) the highest possible filling on each site is $p-1$, thus generalising the usual Pauli principle. Furthermore, we can indeed define the number operator,

$$N_j = \sum_{m=1}^{p-1} F_j^{\dagger m} F_j^m, \tag{12}$$

which obeys the usual algebra with creation and annihilation operators,

$$\left[N_j, F_j^\dagger\right] = F_j^\dagger, \qquad \left[N_j, F_j\right] = -F_j, \tag{13}$$

and acts as follows on the Fock states as

$$N_j |n_1, n_2, \ldots, n_L\rangle = n_j |n_1, n_2, \ldots, n_L\rangle. \tag{14}$$

Finally we note that for $p = 4$ the FPF operators can be linked to spinful fermions via a non-linear relation [33]. However, in our work we will not use this since we focus exclusively on the case $p = 3$ in the following.

# 3 The model and its phase diagram

In this section we introduce the model and its symmetries and present its phase diagram, the main result of this paper. We discuss the observables and correlation functions which will be used to analyse the different phases in Section 5.

Having introduced the operators creating and annihilating FPFs in the previous section, we are now in the position to define the model which we will study in this paper. We restrict ourselves to the simplest non-trivial case of $p = 3$ and consider the one-dimensional Hamiltonian

$$H(g) = -t \sum_{j=1}^{L-1} \left[ (1-g)F_j^\dagger F_{j+1} + g F_j^{\dagger 2} F_{j+1}^2 + \text{h.c.} \right]. \tag{15}$$

Throughout our work we set $t = 1$ and use it as the energy unit. The parameter $g$ is restricted to the interval $0 \le g \le 1$, interpolating between the extreme cases of pure single-particle hopping and pure coherent pair hopping. The latter is allowed due to the possibility of having two FPFs at the same lattice site. We consider a one-dimensional chain of $L$ lattice sites with free boundary conditions.

We note that the three-state quantum Potts chain (1) can in principle also be written in terms of FPFs. However, the resulting expression is much more complicated than the hopping Hamiltonian (15), containing for example terms that break the particle-number conservation.

The model (15) with $g = 0$, ie, the case of pure single-particle hopping, was studied by Rossini et al. [31]. They showed that there exists a Mott-like phase at unit filling, ie, if there are $L$ FPFs in total, while at all other filling fractions the model is gapless and can be described by an anyonic Luttinger liquid [32]. The aim of our work is to extend the analysis to $g \ne 0$ and study the effect of the additional pair hopping on the phase diagram.

Coming back to the Hamiltonian (15), we observe a U(1) symmetry which results in the conservation of the total number of particles, $N = \sum_{j=1}^{L} N_j$, as can be checked using Equation (13). Moreover the model is invariant under the particle-hole transformation $F_j \to F_j^\dagger$. The proof is presented in Appendix A. This implies that, although the Hilbert space can have states with at most $N = 2L$ particles, it is sufficient to restrict the study to those with $N \le L$. Since we are interested in the thermodynamic limit, the relevant quantity would rather be the density or the filling defined by $n = N/L$. Therefore we will present the results for $0 < n \le 1$.

Our main result is the phase diagram of the model (15) which is presented in Figure 1. The phase diagram consists of four phases: the left phase (white region in Figure 1, which will be indicated by L throughout the paper), the right phase (yellow region, indicated by R), the middle phase (orange region, indicated by M) and the gapped phase (the thick violet line at $n = 1$, indicated by G). To characterise and distinguish different phases we look into different properties and observables: the energy gap, the entanglement entropy and two-point correlation functions. The results of this characterisation are summarised in Table 1: we find two gapless phases (L and R) that allow a Luttinger liquid description ($c = 1$) which are distinguished by the different power-law behaviour of the correlation functions, another gapless phase (M) with central charge $c = 2$, and a gapped phase (G) which can be regarded as the extension of the anyonic Mott-like phase to $g \ne 0$. A detailed discussion of the four phases is given in Secion 5.

Studying the energy difference between the ground state and the first excited state, $\delta(L) = E_1(L) - E_0(L)$, as a function of system size, $L$, is a classical way of determining whether the model is gapped. For a gapped system this difference will converge to a finite value while for a gapless system it converges to zero as $L^{-z}$, where $z$ is the dynamical critical exponent. For a gapless system in one dimension which can be described by a conformal field theory (CFT) the dynamical critical exponent is $z = 1$ [27, 28]. The scaling behaviour of entanglement

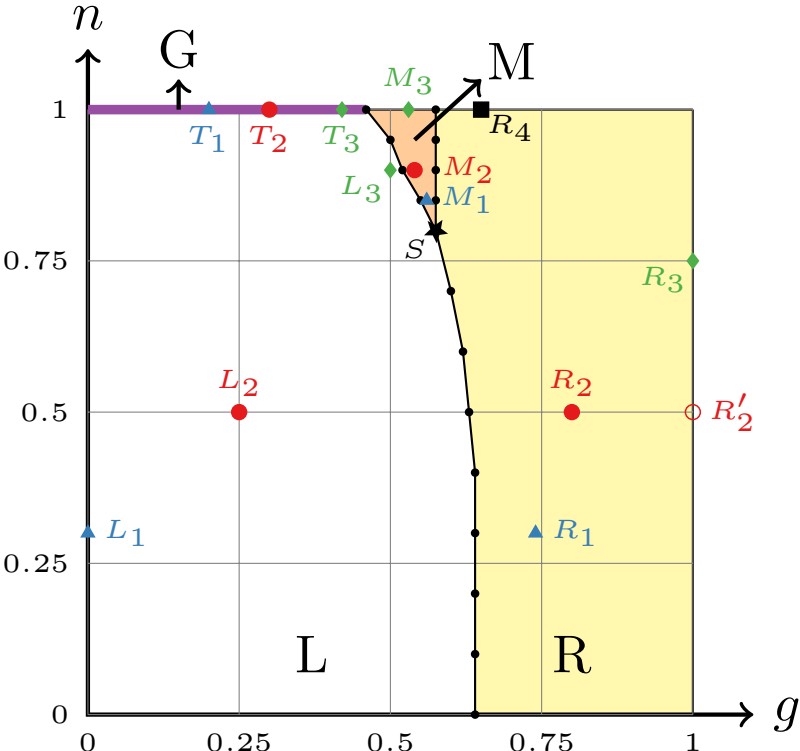

Figure 1: The phase diagram of the model (15). We identified four phases: the left (L) phase (white region), the right (R) phase (yellow), the middle (M) phase (orange) and the gapped (G) phase (thick violet line at unit filling $n = 1$). The properties of the phases are summarised in Table 1. The detailed analyses at the coloured points ($L_{1,2,3}$ etc.) are presented in Section 5. The black star, $S \simeq (0.58, 0.80)$, indicates the point where the three phases, L, R and M, meet. The phase transitions have been determined at the black dots; for fixed $n$ the estimated uncertainty is of the order of $\Delta g = 0.01$. The transition between the L and R phase seems to be second order.

entropy (EE), $S(l)$, as a function of subsystem size, $l$, is another probe to separate different phases from each other. For a gapped phase the EE saturates to a constant value. For a gapless system, however, the EE grows with the subsystem size. For an open chain at criticality with an underlying CFT, one can read off the central charge, $c$, using the Calabrese-Cardy (CC) formula [34, 35],

$$S(l) = \frac{c}{6} \log \left[ \frac{L}{\pi} \sin \left( \frac{\pi l}{L} \right) \right] + S_0, \tag{16}$$

in which $S_0$ is a non-universal constant. Finally, correlation functions play an essential role in our understanding of the phases. In a gapped phase a typical two-point correlation function decays exponentially as a function of distance with a correlation length of the order of the inverse gap. For a gapless system, however, the two-point correlation functions show power-law behaviour. Hence, following Reference [31], we will also study the two-point correlation functions of FPF operators

$$G_1(r) = \left| \left\langle F^\dagger_{\frac{L}{2} - \frac{r}{2}} F_{\frac{L}{2} + \frac{r}{2}} \right\rangle \right|, \qquad G_2(r) = \left| \left\langle F^{\dagger 2}_{\frac{L}{2} - \frac{r}{2}} F^2_{\frac{L}{2} + \frac{r}{2}} \right\rangle \right|. \tag{17}$$

Table 1: Summary of the properties of the four phases in Figure 1. The central charge $c$ is obtained from the fit of the EE to the CC fomula (16). In the L and R phase we obtain the value $c = 1$ up to about 1%. In the M phase the deviation from $c = 2$ is slightly larger, as indicated in the inset of Figure 7(a).

| phase | energy gap | $c$ | $G_1(r)$ | $G_2(r)$ |
|-------|-----------|-----|----------|----------|
| L | gapless | 1 | $\sim r^{-2/3}$ | $\sim r^{-\alpha_2(g,n)}$ |
| R | gapless | 1 | 0 | $\sim r^{-13/18}$ |
| M | gapless | 2 | $\sim r^{-\alpha_1'(g,n)}$ | $\sim r^{-\alpha_2'(g,n)}$ |
| G | gapped | - | $\sim \exp[-r/\xi_1(g)]$ | $\sim \exp[-r/\xi_2(g)]$ |

We measure the correlations between two lattice sites of distance $r$ which are symmetrically distributed around the middle of the chain. This is to minimise the finite-size effects from the edges.

The analysis of the phases using the tools discussed above will be presented in Section 5. In addition, in some cases it is also possible to employ analytical methods like bosonisation [36, 37], which for example allows us to obtain an effective Luttinger liquid description in the L and R phases. Before presenting the detailed results for the phase diagram we will briefly discuss the implementation of our numerical simulations in the next section.

## 4 The implementation for numerical studies

To study the model numerically we performed density matrix renormalisation group (DMRG) simulations [38,39] using the ALPS [40–42] and TeNPy [43,44] libraries and checked that the obtained results are the same. To implement the model for performing DMRG and bosonisation we use the Fradkin–Kadanoff transformation [22],

$$F_j = \left(\prod_{k=1}^{j-1} U_k\right) B_j, \tag{18}$$

where

$$U_k = \mathbf{1} \otimes \cdots \otimes \underbrace{U}_{k} \otimes \cdots \otimes \mathbf{1}, \qquad U = \begin{pmatrix} 1 & 0 & 0 \\ 0 & \omega & 0 \\ 0 & 0 & \omega^2 \end{pmatrix}, \tag{19}$$

$$B_j = \mathbf{1} \otimes \cdots \otimes \underbrace{B}_{j} \otimes \cdots \otimes \mathbf{1}, \qquad B = \begin{pmatrix} 0 & 1 & 0 \\ 0 & 0 & 1 \\ 0 & 0 & 0 \end{pmatrix}. \tag{20}$$

The matrix representations of the local operators $U$ and $B$ are given in the local basis where the clock operator $\tau$ is diagonal, ie, $U = \tau$. The operators acting on different sites commute while the on-site algebra is given by

$$B_j U_j = \omega U_j B_j. \tag{21}$$

Applying this transformation together with the resulting relation $B_j^{\dagger 2} U_j^2 = B_j^{\dagger 2}$, the Hamiltonian (15) becomes

$$H(g) = -t \sum_{j=1}^{L-1} \left[ (1-g) B_j^\dagger U_j B_{j+1} + g B_j^{\dagger 2} B_{j+1}^2 + \text{h.c.} \right],\qquad(22)$$

which is local and only consists of bosonic degrees of freedom. Hence it can be easily implemented for the DMRG calculation.

The DMRG simulations for the entanglement entropy and the correlation functions were performed for a chain of size $L = 240$, our default system size. To find the central charge of the gapless phases using the CC formula or its modified variation, as it will be later introduced, we dropped the first and the last ten sites to stay away form finite-size effects due to the edges. The data for the correlation functions will be presented for $r \in [10 - L/2]$ and the same interval will be used for the fittings. For the finite-size scaling of the energy gap we use a range of system sizes, usually between $L = 64$ and $L = 240$. The DMRG was performed with the bond dimension $\chi = 500$ in the L, R and G phases, and $\chi = 800 - 1000$ in the M phase. The number of sweeps which is needed for the convergence varies and depends on the parameters. The typical number of sweeps in the L, R and G phases is between 20 and 50. In the M phase, however, 40 to 60 sweeps were done. Each sweep consists of minimisation from the first site to the very last one and then from the last site back to the first one.

## 5 The results

In this section we present the detailed results of our numerical and analytical study of the phase diagram. The specific values of the parameters at which we present numerical data are indicated by coloured points in Figure 1. We will use the same colour to present the EE and correlation functions $G_1$ and $G_2$ for each one of these points.

### 5.1 The L phase

Rossini et al. [31] studied the model (15) for the special case of $g = 0$ and various filling fractions $n$. They found that the model is gapless for any filling $n < 1$ and well described by an anyonic Luttinger liquid [32] with Luttinger parameter $K = p/2$ such that the correlation functions decay as power laws $G_1(r) \sim r^{-\alpha_1}$ with $\alpha_1 = 2/p$ and $G_2(r) \sim r^{-\alpha_2}$ with $\alpha_2 = 4\alpha_1$. Although the numerical results of Reference [31] match very well with the theoretical predictions derived by Calabrese and Mintchev [32] for $G_1$, there are discrepancies between the theory and the numerics for $G_2$. Our numerical and analytical results show that the properties of the model at $g = 0$ extend to a finite region with $g > 0$.

The results of the numerical calculations in the L phase are shown at the points $L_1 = (g, n) = (0, 0.3)$, $L_2 = (0.25, 0.5)$ and $L_3 = (0.5, 0.9)$. These points were selected to show the typical behaviour. The L phase, which is depicted as a white region in Figure 1, is found to be gapless with the central charge $c = 1$, as is confirmed by the fit of the EE shown in Figure 2(a) to the CC formula. In Figure 2(b) we show the energy difference $\delta(L) = E_1(L) - E_0(L)$ at the point $L_2$ and system sizes $L \in [64 - 176]$. We used a power-law function for the fitting, $\delta(L) = a/L^b + \delta_0$, which gave us $b \approx 0.99$ and $\delta_0 \simeq 10^{-4}$. Therefore we can conclude that the dynamical critical exponent is given by $z = 1$, which confirms that the low-energy physics can be described by a CFT.

In Figure 3 we present the two-point correlation functions $G_1(r)$ and $G_2(r)$ for the same three points in the L phase. The correlation function $G_1(r)$ shows a power-law behaviour, $G_1(r) \sim r^{-\alpha_1}$ with $\alpha_1 \approx 2/3$, as it was the case for $g = 0$. In addition we observe weak

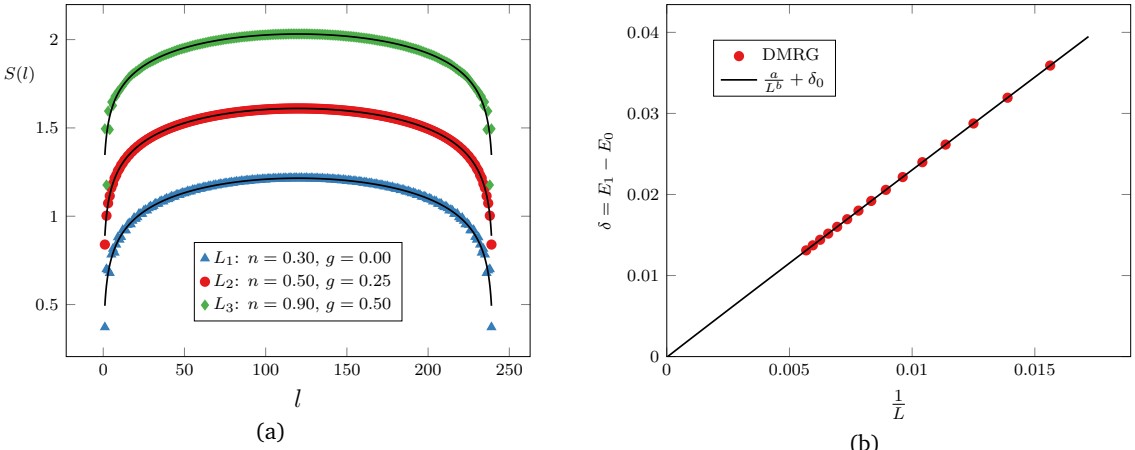

Figure 2: EE and gap for the points $L_1$, $L_2$ and $L_3$ in the L phase. (a) EE as a function of subsystem size $l$ for a chain of size $L = 240$. The solid lines are the CC formula with $c = 1$. We have shifted the red points by 0.2 and the green points by 0.4 for visibility. (b) The energy difference between the first excited state and the ground state at the point $L_2$ as a function of $1/L$ for $L \in [64 - 176]$. The fitting parameters for the solid line are $b \approx 0.99$ and $\delta_0 \simeq 10^{-4}$, thus indicating a gapless phase.

oscillations with a wave number $q_1$ that takes the values $q_1 \approx 0.95$ at $L_1$ and $q_1 \approx 1.57$ at $L_2$, while at $L_3$ we were not able to determine $q_1$ with sufficient accuracy. The origin of these oscillations seems to involve doubly-occupied sites, as is indicated by comparison to the bosonisation treatment (see below). The result on the correlation function $G_2$ shows a power-law decay too, $G_2(r) \sim r^{-\alpha_2}$, but the exponent $\alpha_2$ depends on both the pairwise hopping, $g$, and the filling fraction, $n$, as it is indicated in the inset.

In the following we provide an argument for our finding of $G_1(r) \sim r^{-2/3}$ based on a bosonisation [36, 37] treatment. Or starting point is the observation that the probability to have two particles at the same site is strongly suppressed throughout the L phase. For example, at the point $L_2$ the probability of having an empty site, a site with one particle and a site with two particles are $P(0) \simeq 0.54$, $P(1) \simeq 0.42$ and $P(2) \simeq 0.04$, respectively. Therefore one can argue that it is reasonable to project the model to the local Hilbert space with at most one particle at a given site, and thus drop the second term in the Hamiltonian. Using this projection, we can identify the operator $B_j$ in the subspace spanned by $|0\rangle$ and $|1\rangle$ with the raising spin-1/2 operator $\sigma_j^+$,

$$B_j \rightarrow \sigma_j^+ = \begin{pmatrix} 0 & 1 \\ 0 & 0 \end{pmatrix}, \tag{23}$$

and simplify $G_1(r)$ to

$$G_1(r) = \left| \left\langle F_0^\dagger F_r \right\rangle \right| = \left| \left\langle B_0^\dagger U_0 U_1 \cdots U_{r-1} B_r \right\rangle \right| \sim \left| \left\langle \sigma_0^- U_0^{(p)} U_1^{(p)} \cdots U_{r-1}^{(p)} \sigma_r^+ \right\rangle \right|, \tag{24}$$

in which

$$U_k^{(p)} = \mathbf{1} \otimes \cdots \otimes \underbrace{U^{(p)}}_{k} \otimes \cdots \otimes \mathbf{1}, \qquad U^{(p)} = \begin{pmatrix} 1 & 0 \\ 0 & \omega \end{pmatrix}. \tag{25}$$

Due to the projection we are left with two states per site. We can use a Jordan–Wigner (JW) transformation and relate the spin-1/2 operators to a set of spinless fermions, $\psi_j$,

$$\sigma_j^z = 2n_j - 1, \qquad \sigma_j^+ = e^{i\pi \sum_{k<j} n_k} \psi_j^\dagger, \qquad n_j = \psi_j^\dagger \psi_j, \tag{26}$$

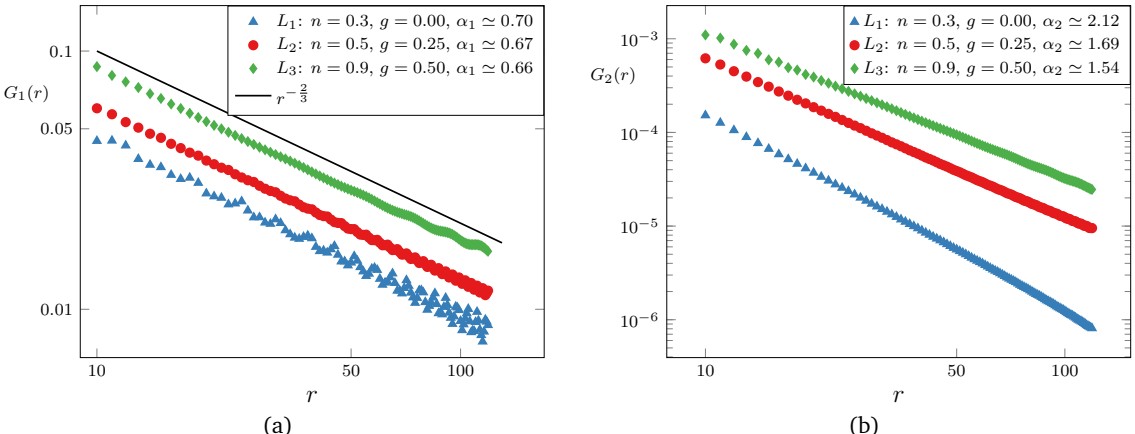

Figure 3: Correlation functions at the three points $L_1$, $L_2$ and $L_3$ in the L phase. (a) $G_1(r)$ as function of $r$. To avoid mixing the data points, we multiplied $G_1(r)$ for the point $L_3$ by 1.5. For comparison $r^{-2/3}$ as derived in (34) is also plotted. (b) $G_2(r)$ as function of $r$. We fitted a power law with exponent $\alpha_2$, the obtained values are given in the legend. We note that both correlation functions show a power-law decay, and that $G_2(r) \ll G_1(r)$ at small $r$ consistent with the strongly suppressed probability to find two particles at the same site.

where the $\psi_j$ satisfy $\{\psi_i, \psi_j\} = 0$ and $\{\psi_i, \psi_j^\dagger\} = \delta_{ij}$. Applying the JW transformation to $G_1(r)$ and rewriting the matrix $U^{(p)}$ we obtain

$$G_1(r) \sim \left| \left\langle \sigma_0^- U_0^{(p)} U_1^{(p)} \cdots U_{r-1}^{(p)} \sigma_r^+ \right\rangle \right| = \left| \left\langle \sigma_0^- \left[ \prod_{k=0}^{r-1} e^{i\frac{2\pi}{3}(1-n_k)} \right] \sigma_r^+ \right\rangle \right| \tag{27}$$

$$= \left| \left\langle \psi_0 \, e^{-i\frac{2\pi}{3}\sum_{k=0}^{r-1} n_k} e^{i\pi \sum_{l=0}^{r-1} n_l} \psi_r^\dagger \right\rangle \right| = \left| \left\langle \psi_0 \, e^{i\frac{\pi}{3}\sum_{k=0}^{r-1} n_k} \psi_r^\dagger \right\rangle \right|. \tag{28}$$

Assuming that the fermions have a Fermi surface, we can linearise around the two resulting Fermi points $k = \pm k_F$,

$$\psi_j = \sqrt{a} \left[ e^{ik_F x} \psi_+(x) + e^{-ik_F x} \psi_-(x) \right], \tag{29}$$

where $a$ denotes the lattice constant and $x = ja$ the spatial coordinate that will be treated as a continuous variable. In addition we use the bosonisation dictionary [36, 37],

$$\psi_\pm(x) = \frac{1}{\sqrt{2\pi\alpha}} e^{i\sqrt{\pi}[\pm\phi(x)-\theta(x)]}, \tag{30}$$

in which $\alpha^{-1}$ is the momentum cut-off, and $\phi(x)$ and $\theta(x)$ are dual fields that satisfy the commutation relation $[\phi(x), \theta(y)] = i\Theta(y-x)$, with $\Theta(x)$ being the Heaviside step function. To continue we recall that for bosonisation normal ordering is necessary. Hence for the density operator we use $n_k =: n_k : +\bar{n}$, in which $\bar{n}$ is the average density on each site in the ground state and $: n_k := \partial_x \phi/\sqrt{\pi}$. Furthermore, assuming that the interactions are incorporated via a Luttinger parameter $K$ we rescale the bosonic fields as $\phi(x) \to \sqrt{K}\phi(x)$, $\theta(x) \to \theta(x)/\sqrt{K}$ to bring the correlation function into the standard form

$$G_1(r) \sim \left| \left\langle \left[ e^{i\sqrt{\pi}\left[\sqrt{K}\phi(0) - \frac{\theta(0)}{\sqrt{K}}\right]} + e^{-i\sqrt{\pi}\left[\sqrt{K}\phi(0) + \frac{\theta(0)}{\sqrt{K}}\right]} \right] e^{i\frac{\sqrt{\pi K}}{3}[\phi(r)-\phi(0)]} \right. \right.$$
$$\left. \left. \times \left[ e^{-i\sqrt{\pi}\left[\sqrt{K}\phi(r) - \frac{\theta(r)}{\sqrt{K}}\right]} e^{-ik_F r} + e^{i\sqrt{\pi}\left[\sqrt{K}\phi(r) + \frac{\theta(r)}{\sqrt{K}}\right]} e^{ik_F r} \right] \right\rangle \right|. \tag{31}$$

Using the Wick theorem, the neutrality condition for vertex operators, and

$$\left\langle e^{i\beta[\phi(r)-\phi(0)]}\right\rangle = \left\langle e^{i\beta[\theta(r)-\theta(0)]}\right\rangle = \left(\frac{\alpha^2}{\alpha^2+r^2}\right)^{\frac{\beta^2}{4\pi}} \tag{32}$$

we get

$$G_1(r) \sim A_1 \left(\frac{1}{r}\right)^{\frac{1}{2K}+\frac{2}{9}K}\left[1+\cos(2k_\mathrm{F}r)\left(\frac{\alpha}{r}\right)^{\frac{2}{3}K}\right], \tag{33}$$

where we have limited ourselves to the two leading terms at large separations, and $A_1$ is a non-universal constant. For $K > 0$ the first term in $G_1(r)$ decays slower than the second one and thus is dominant at large separations. Hence we conclude that at large $r$

$$G_1(r) \sim r^{-\frac{1}{2K}-\frac{2}{9}K} \sim r^{-\frac{2}{3}}, \tag{34}$$

where in the last step we have used that for the free anyon gas [31,32] the Luttinger parameter $K$ is related to the statistical parameter $\kappa = \theta/\pi$ via $K = 1/\kappa = 3/2$. We stress that we have derived the result (34) from the microscopic model (15), thereby linking it to the phenomenological theory applied by Calabrese and Mintchev [32]. In particular, our line of argument shows why the anyonic Luttinger model indeed provides a good description of the L phase. We note, however, that the oscillations in $G_1(r)$ observed in Figure 3(a) are not adequately described by the second term in (33). Thus they are not captured by the line of argument presented above, which hints at the importance of doubly-occupied sites. Moreover, the behaviour $G_2(r) \sim r^{-\alpha_2}$ cannot be described by the bosonisation approach, as obviously doubly occupancy will be relevant for this correlation function. We do not have a clear understanding yet how the oscillations in $G_1(r)$ or the power-law scaling of $G_2(r)$, in particular the exponent $\alpha_2$, relate to the filling $n$ and the parameter $g$.

## 5.2 The R phase

The parameter $g$ controls the relative strength of single-particle and pair-hopping amplitudes. By increasing $g$ for the filling $n \lesssim 0.8$ the system directly enters the R phase (yellow region in Figure 1) from the L phase. For larger filling, $0.8 \lesssim n < 1$, there exists a phase with the central charge $c \approx 2$ between the L phase and the R phase. This M phase will be discussed in Section 5.3. The point where the three phases L, R and M meet is located at $S \simeq (0.58, 0.80)$ and marked with a black star in the phase diagram.

In this section we present details on the R phase. Numerical results are shown for the selected points $R_1 = (0.74, 0.3)$, $R_2 = (0.8, 0.5)$ and $R_3 = (1, 0.75)$. The EE and energy gap at these three points are given in Figure 4. We conclude that also the R phase is gapless with central charge $c = 1$. More precisely, the energy gap scales as $\delta(L) = a/L^b + \delta_0$ with $b \approx 0.99$ and $\delta_0 = 10^{-4}$, thus the dynamical critical exponent is given by $z = 1$.

The difference between the L and R phases shows up only when considering the correlation functions. In the R phase the correlation function $G_1(r)$ decays exponentially as a function of distance $r$, $G_1(r) \sim \exp(-r/\xi_1)$, with a correlation length, $\xi_1$, of the order of a few lattice constants. Away from the phase transition one even finds $\xi_1 \sim a$, ie, the correlation function essentially vanishes. This finding can be understood by noting that deep in the R phase the probability of having one particle on a site is generally much smaller than having two particles or an empty site. For instance, at the point $R_2$ the probability of having an empty site, a site with one particle and a site with two particles are $P(0) \simeq 0.24$, $P(1) \simeq 0.01$ and $P(2) \simeq 0.75$, respectively. In the special case of $g = 1$ we even find $P(1) = 0$ in the ground state. This can be understood from the Hamiltonian $H(1)$, in which only the operators $F_j^2$ or $F_j^{\dagger 2}$ appear, and

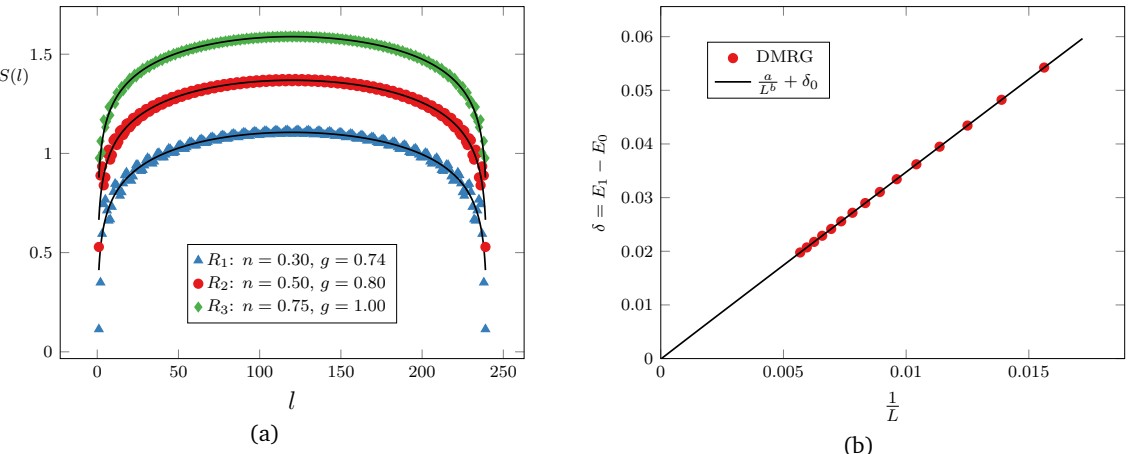

Figure 4: EE and energy gap for three points $R_1$, $R_2$ and $R_3$ in the R phase. (a) EE as a function of subsystem size $l$. The solid lines are the CC formula with $c = 1$. We have shifted the red points by 0.2 and the green points by 0.4 for visibility. (b) Energy gap above the ground state at the point $R_2$. The fitting parameters for the solid line are $b \approx 0.99$ and $\delta_0 \simeq 10^{-4}$, again indicating a gapless phase.

both annihilate the one-particle state. So the on-site one-particle sector decouples and does not play a crucial role on the low-energy physics.

We can use this information from the numerics and assume that in the R phase the low-energy physics can be captured by the second term in the Hamiltonian only, ie, we approximate

$$H_R^{(p)}(g) = -tg \sum_{j=1}^{L-1} F_j^{\dagger 2} F_{j+1}^2 + \text{h.c.} = -tg \sum_{j=1}^{L-1} B_j^{\dagger 2} B_{j+1}^2 + \text{h.c..} \tag{35}$$

Following our line of argument used above for the L phase we project the Hamiltonian onto the space with empty or doubly occupied on-site subspaces $|0\rangle$ and $|2\rangle$, respectively. Hence we can identify the operator $B_j^2$ with the raising spin-1/2 operator $\sigma_j^+$ in this subspace,

$$B_j^2 \to \sigma_j^+ = \begin{pmatrix} 0 & 1 \\ 0 & 0 \end{pmatrix}, \tag{36}$$

which gives rise to the XX-Hamiltonian,

$$H_R^{(p)}(g) = -tg \sum_{j=1}^{L-1} \sigma_j^- \sigma_{j+1}^+ + \text{h.c..} \tag{37}$$

This projected Hamiltonian is quite fruitful. First of all we note that it is well-known that the XX-model is gapless and can be described with the bosonic CFT with the central charge $c = 1$ [27, 28]. Moreover, we can calculate $G_2(r)$ in the same way that we calculated $G_1(r)$ in the L phase,

$$G_2(r) = \left| \langle F_0^{\dagger 2} F_r^2 \rangle \right| = \left| \langle B_0^{\dagger 2} U_0^2 U_1^2 \cdots U_{r-1}^2 B_r^2 \rangle \right|. \tag{38}$$

Using the definition of the matrix $U$, we see that the projection of $U^2$ onto the subspace spanned by $|0\rangle$ and $|2\rangle$ has the same form as the matrix $U^{(p)}$ in Equation (25). Therefore the calculation we presented for the correlation function $G_1(r)$ in Section 5.1 can be directly applied to the correlation function $G_2(r)$ in the R phase. Furthermore, since the XX-model is a free theory

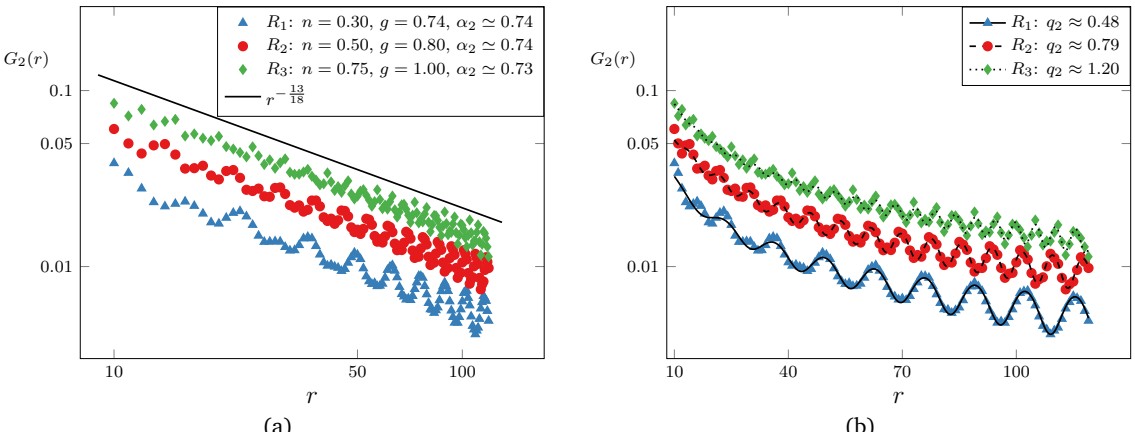

Figure 5: The correlation function $G_2(r)$ is plotted as function of $r$ for the three points $R_1$, $R_2$ and $R_3$ in the R phase. To avoid mixing the data points, we multiplied $G_2(r)$ by 1.5 and 2 for the red and green data points, respectively. (a) For comparison we plot the prediction (39) as solid line. (b) We fitted the data with sub-leading oscillations decaying as a power law, ie, $G_2(r) = A_2 r^{-\frac{13}{18}} + A'_2 r^{-\beta_2} \cos(q_2 r + \phi_2)$. The resulting wave numbers $q_2$ are given in the legend. For the point $R_3$ the wave length $2\pi/q_2 \approx 5$ becomes rather short, increasing the uncertainty in the fit. The accuracy of the fit for the exponent $\beta_2$ was not sufficient to obtain reliable results.

it seems reasonable to set the Luttinger parameter to its non-interacting value, $K = 1$. As a result we finally arrive at the prediction

$$G_2(r) \sim r^{-\frac{1}{2}-\frac{2}{9}} = r^{-\frac{13}{18}}. \tag{39}$$

In Figure 5 we present the correlation function $G_2(r)$ for the three points $R_{1,2,3}$ deep in the R phase. The agreement between the numerical results and the simple prediction (39) from LL theory is quite good. On top of the power-law decay we observe oscillations with a wave number $q_2$. As can be seen from the fitted values given in the legend of Figure 5(b), the wave number strongly depends on the filling fraction $n$. On the other hand, we determined the wave number at the point $R'_2 = (1, 0.5)$ to be $q_2 \approx 0.8$, indicating that there seems to be no (strong) dependence on the parameter $g$. This is also consistent with results obtained along the cut $(g, 0.3)$ for $0.6 \leq g \leq 0.7$ (not shown, see Figure 12 for the energy along this cut) which show essentially constant wave numbers for both $G_1(r)$ and $G_2(r)$ within the phases L and R.[4] Furthermore, the oscillations seem not to be described by the first correction to (39), ie, they are not captured by the Luttinger liquid description of $G_2(r)$. Thus at the moment we lack a clear understanding of the oscillations.

Finally we note that there are subtleties in the R phase at the filling $n = 1$. In Figure 6 we present the EE and the pair correlation function $G_2(r)$ for the point $R_4 = (0.65, 1)$. The correlation function $G_1(r)$ vanishes, as it is the case throughout the R phase. Due to the bifurcation in the EE profile, in order to find the central charge we use the modified CC formula [45, 46],

$$S(l) = \frac{c}{6} \log\left[\frac{L}{\pi} \sin\left(\frac{\pi l}{L}\right)\right] + S_0 + \frac{a_1 + a_2 \cos(\pi l)}{\left[\frac{L}{\pi} \sin\left(\frac{\pi l}{L}\right)\right]^b}, \tag{40}$$

---

[4]Incidentally we observe that for a fixed filling fraction $n$ the wave numbers are approximately related by $q_1 \approx 2q_2$, both at $n = 0.3$ and $n = 0.5$.

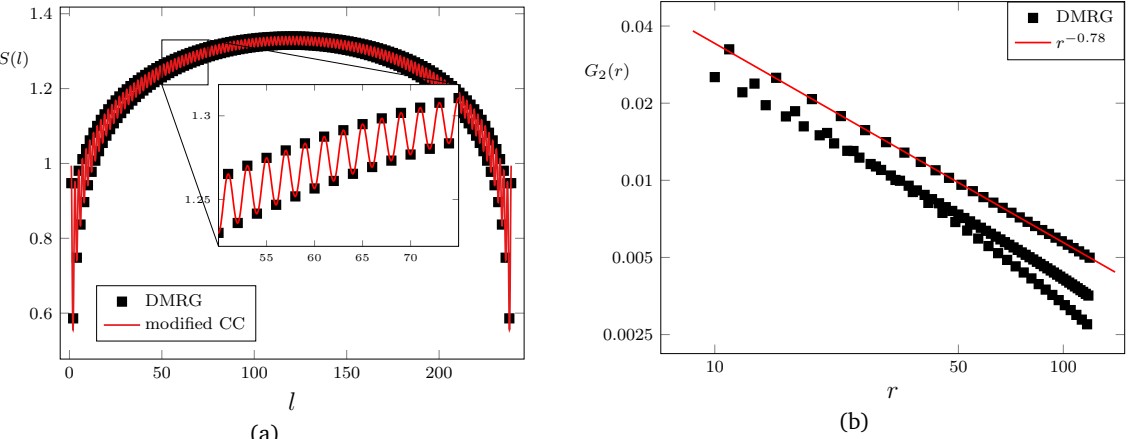

(a)

(b)

Figure 6: EE and the correlation function $G_2(r)$ at the point $R_4 = (0.65, 1)$. (a) The EE as a function of subsystem size $l$ together with a fit of the modified CC formula (40). We find $c = 1$ and $b = 0.78$. The inset shows the bifurcation of the data points between even and odd $l$. (b) The correlation function $G_2(r)$ together with a fit (red solid line) to the upper branch of the data.

in which $a_1$, $a_2$ and $b$ are new fitting parameters in addition to the central charge $c$ and the constant $S_0$. Using the modified CC formula we get the central charge $c = 1$ for the filling $n = 1$ in the R phase, just as was obtained for lower fillings. The same bifurcation also appears in the correlation function $G_2(r)$. Therefore, in order to extract a power law we picked the upper part of the data for fitting with the result $G_2(r) \sim r^{-0.78}$, which is still quite close to the prediction $13/18 \approx 0.72$ we obtained deep in the R phase from bosonisation. The difference between the prediction and the numerical value could be due to the fit to the upper part of data and the fact that at this point $P(1) \simeq 0.1$, which means that the local state $|1\rangle$ plays a more important role than it does deep in the R phase.

## 5.3 The M phase

For sufficiently large filling fractions, $0.8 < n \leq 1$, another gapless phase between the L and R phases exists. This M phase is indicated as the orange region in the phase diagram, Figure 1. The M phase is found to be gapless with central charge $c = 2$, as can be deduced from the fit of the CC formula (16) to the EE calculated at the points $M_1 = (0.56, 0.85)$, $M_2 = (0.54, 0.9)$ and $M_3 = (0.53, 1)$ shown in Figure 7(a). Verifying the CFT prediction regarding the scaling of the low-lying energy levels, $\delta(L) \sim 1/L$, turned out to be a hard task. This could be due to two issues: The M phase is a fairly small region, therefore any chosen point is quite close to the phase boundaries with the L and the R phases. This in turn demands very large system sizes. In addition, the high central charge $c = 2$ and oscillatory features suggest that larger bond dimensions are required. In Figure 7(b) we present our results for the energy gap at the point $M_3$, system sizes $L \in [16 - 120]$ and bond dimension $\chi = 1000$. While we observe a strongly fluctuating dependence on the system size, the results clearly indicate a vanishing of the energy gap in the thermodynamic limit.

The two-point correlation functions $G_1(r)$ and $G_2(r)$ are presented in Figure 8. They both show a power-law behaviour as it is expected from CFT. The correlation function $G_1(r)$ is quite smooth and behaves as $G_1(r) \sim r^{-\alpha_1}$ with an exponent $\alpha_1 \simeq 0.75 - 0.8$. Although ripples and fluctuations in the correlation functions $G_2(r)$ are clearly visible, it still has a power-law trend, $G_2(r) \sim r^{-\alpha_2}$ with $\alpha_2 \simeq 1.1$. Since in the M phase all three states at each site play a role, it is

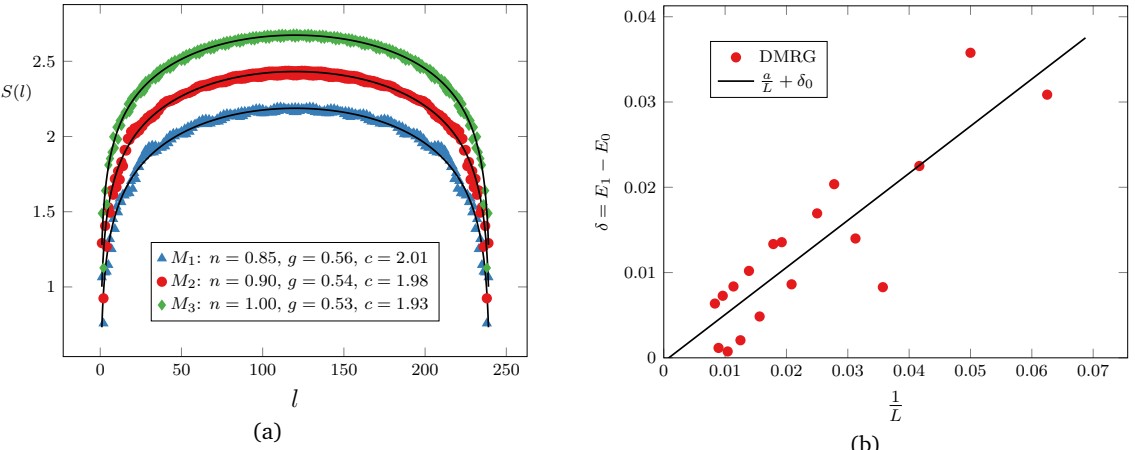

Figure 7: (a) EE as a function of subsystem size $l$ for the points $M_1$, $M_2$ and $M_3$ in the M phase. The fits are performed with the CC formula (16), giving a central charge of $c \approx 2$. We have shifted the red points by 0.2 and the green points by 0.4 for visibility. (b) Energy gap above the ground state at the point $M_3$. The fitting parameters for the solid line are $b \approx 1$ and $\delta_0 \simeq 10^{-4}$.

not clear at this point whether one can relate the properties of this phase to a Luttinger liquid picture.

The location of the M phase between the L and R phases suggests the following interpretation: In the M phase one has two sets of gapless bosonic modes, which is supported by its central charge $c = 1 + 1 = 2$. A priori we do not see a reason why these two theories should have the same effective velocity.[5] Now, when crossing the phase boundary to the L phase, a gap opens in one of the bosonic theories (which is naively related to pair excitations), while when going to the R phase the other theory develops a gap (naively related to single-particle excitations).

## 5.4 The G phase

Finally we consider the gapped G phase indicated by a thick violet line in Figure 1. This phase was identified by Rossini et al. [31] at $g = 0$ and interpreted as an anyonic Mott-like phase. Our analysis reveals that this phase extends to finite values of $g$ with the transition to the gapless M phase located at $g \simeq 0.45$. Using DMRG we numerically calculated the energy gap as a function of system size, $\Delta(L)$, and used a power-law fit to extract the gap $\Delta = \Delta(g)$ in the thermodynamic limit via

$$\Delta(L) = \frac{a}{L^b} + \Delta. \tag{41}$$

The finite-size data and fits as well as the $g$-dependence of the extracted gap $\Delta(g)$ are presented in Figure 9. For convenience we rescaled the gap with its value at $g = 0$, namely $\Delta(0) = 0.106\,t$.

We have calculated the EE and correlation functions at the points $T_1 = (0.2, 1)$, $T_2 = (0.3, 1)$ and $T_3 = (0.42, 1)$ in the G phase. The data are shown in Figures 10 and 11, respectively. The EE saturates quite quickly as a function of subsystem size $l$ to a constant value, which is indicative of a finite correlation length [34]. This is also supported by the behaviour of the

---

[5]The situation is reminiscent to the one-dimensional Hubbard model away from half filling [47], and might be similar to the $c = 3/2$ phase recently discussed in Reference [48].

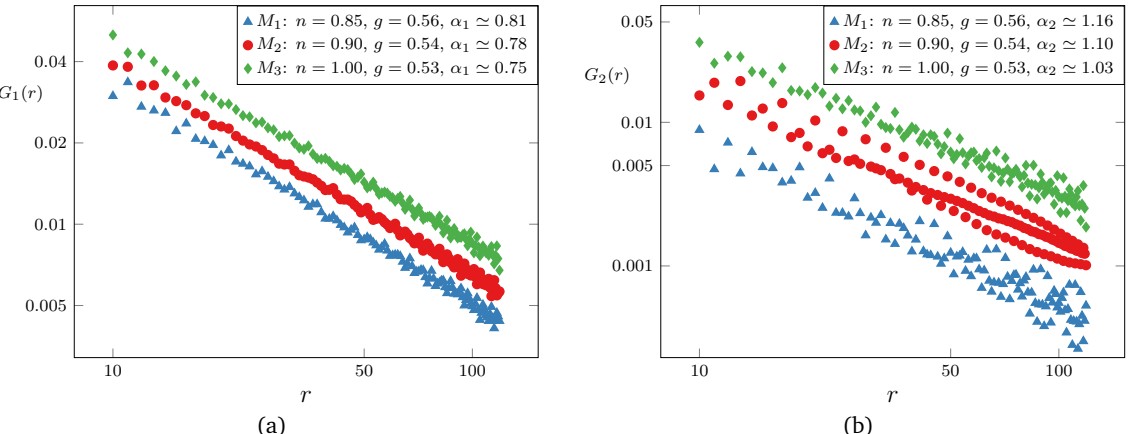

Figure 8: The correlation functions $G_1(r)$ and $G_2(r)$ for the three points, $M_1$, $M_2$ and $M_3$, in the M phase are plotted in (a) and (b), respectively. To avoid mixing the data points, $G_1(r)$ for the point $M_3$ was multiplied by 1.2, $G_2(r)$ for $M_2$ was multiplied by 2.5 and $G_2(r)$ for $M_3$ was multiplied by 3.2. For both correlation functions we fitted power laws with exponents $\alpha_{1,2}$, the obtained values are given in the legends.

correlation functions, which show an exponential decay with power-law corrections,

$$G_i(r) = A_i r^{-\beta_i} \exp(-r/\xi_i), \quad i = 1, 2. \tag{42}$$

The obtained fitting parameters are given in Figure 11. The correlations lengths are much smaller than system size, usually of the order 10-20 lattice constants.

## 5.5 On the nature of the transitions

So far we focussed on the properties of the individual phases. In this section we will examine the nature of the transitions between them by studying the ground-state energy and its derivatives together with the information we gathered so far.

### 5.5.1 The transition between the L and the R phases

First we consider the phase transition between the two gapless phases with the central charge $c = 1$, namely the L phase and the R phase. As discussed above, these two phases are best distinguished by the behaviour of the correlation functions and in particular by the vanishing of $G_1(r)$ in the R phase. To further investigate the nature of the transition we calculated the ground-state energy $E(g)$ at a fixed filling. For example, in Figure 12 we show $E(g)$ and its first and second derivatives with respect to $g$ at the filling $n = 0.3$. We see that while the energy and its first derivative are smooth and continuous, there exists a divergence in the second derivative $\frac{\partial^2 E}{\partial g^2}$ at $g_c \simeq 0.64$. This value is identical to the one extracted from the change of the behaviour in $G_1(r)$. We have checked the presence of the two phases down to the filling $n = 0.1$. The transition parameter $g_c(n) = 0.64$ is the same within the accuracy of our numerics for the fillings $0.1 \leq n \leq 0.4$, therefore in Figure 1 we extrapolate it down to $n = 0$. In summary, we conclude that the L and R phases are separated by a phase transition that seems to be of

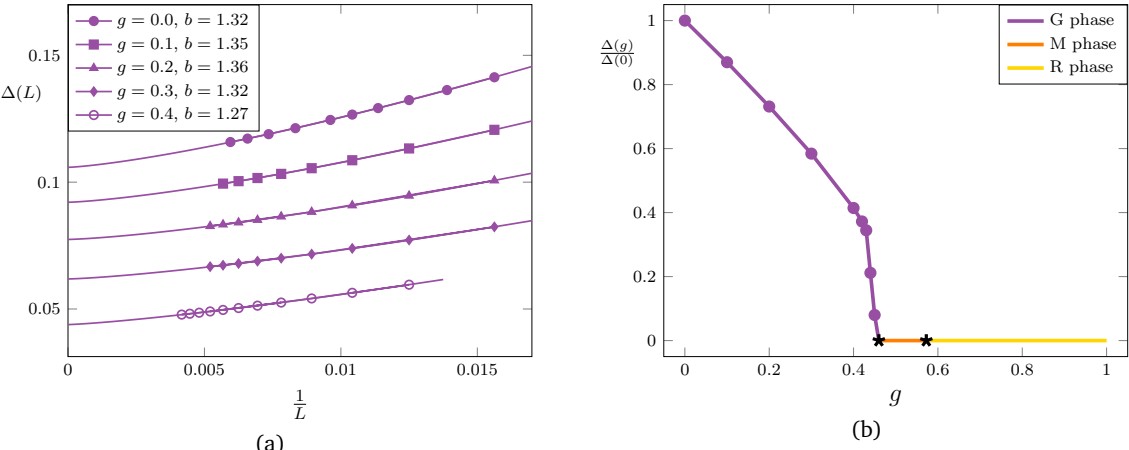

Figure 9: (a) Finite-size scaling of the energy gap, $\Delta(L)$, as a function of system size $L \in [64-240]$ at several points in the G phase. (b) Rescaled energy gap in the thermodynamic limit, $\Delta(g)/\Delta(0)$, as a function of $g$. The orange and the yellow lines correspond to the M and the R phases, respectively, while the stars indicate the transition points.

second order, but that future work is required to obtain a complete characterisation.

### 5.5.2 The transitions to the M phase

For the transition between the M phase and the L and R phases, we studied again the ground-state energy and its first and second derivatives (not shown). While the energy and its first derivative are smooth within our precision, the second order derivative is smooth in the L and the R phases but quite fluctuating and spiky within the M phase. This may be related to the presence of fluctuations as it was recently observed in the incommensurate phase of the Kitaev–Hubbard model [49].

For the phase transitions at $n = 1$ between the M phase and G phase we performed a scaling analysis. For systems of size $L \in [64-100]$ we numerically calculated the energy difference between the first excited state and the ground state, $\Delta(L, g) = E_1(L, g) - E_0(L, g)$. As it is presented in Figure 13(a) the quantity $L^z \Delta$ with $z = 1$ for various system sizes cross at $g_c \simeq 0.45$. This value is consistent with the critical parameter $g_c$ obtained from the EE. Figure 13(b) also shows that by scaling the $g$-axis as $L^{1/\nu}(g - g_c)$ with $\nu = 1$ all the data close to the transition collapse to a single curve. Thus we conclude that our results are consistent with the existence of a second-order transition. We note, however, that reasonable scaling collapse of the data is still obtained if $z$ is varied provided $\nu$ is adapted appropriately.

## 6 Conclusion and outlook

In this work we studied a one-dimensional model for FPFs with $p = 3$, which contained single-particle and coherent pair-hopping terms between nearest-neighbour sites. Using a combination of numerical simulations and analytical arguments we determined the phase diagram as a function of the relative strength between the two hopping terms and the filling fraction, ie, the number of FPFs per lattice site. We identified four different phases: two distinct gapless Luttinger phases with central charge $c = 1$, one gapless phase with $c = 2$, and one gapped phase. All phases were characterised by the energy gap, entanglement entropy and behaviour

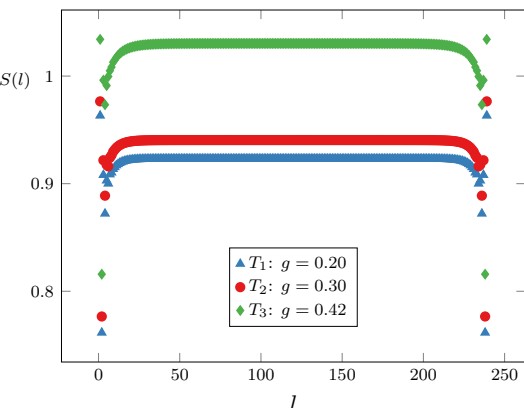

Figure 10: EE as a function of the subsystem size $l$ for three points $T_1$, $T_2$ and $T_3$ in the G phase. We note that in the middle of the chain the EE takes a constant value indicating a finite correlation length [34].

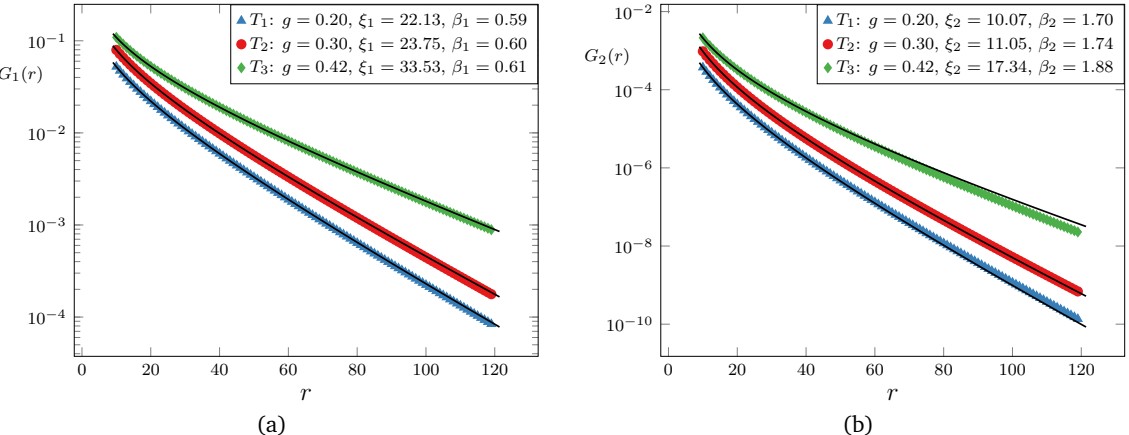

Figure 11: The correlation functions $G_1(r)$ and $G_2(r)$ for the three points $T_1$, $T_2$ and $T_3$ in the G phase are plotted in (a) and (b), respectively. To avoid mixing the data points, $G_1(r)$ for $T_2$ was multiplied by 1.5, $G_1(r)$ for $T_3$ was multiplied by 2, $G_2(r)$ for $T_2$ was multiplied by 2.5 and $G_2(r)$ for $T_3$ was multiplied by 4.5. We note that both correlation functions show an exponential decay at large distances, as indicated by the fitted functions (42) shown as solid lines.

of two-point correlation functions. While we were able to locate the phase transitions accurately, their complete characterisation had to be left for future studies.

Our work can be seen as a step towards the general understanding of the many-particle states of FPFs, or more broadly towards a better understanding of the manifestations of anyonic statistics in many-particle phases. Of course there are many open directions for future research: First, it would be very interesting to analyse the effects of extensions to the simple model (15), for example by including additional complex phases. These are known [50, 51] to have drastic effects on the phase diagram of parafermionic models, and can be crucial for the existence of edge zero modes [25]. Similarly, the addition of BCS-like terms will break the particle number conservation and thus is expected to support additional phases in the phase diagram. Second, studying the properties of FPFs with $p > 3$ is of interest. So far only the pure hopping model (ie, $g = 0$) for $p = 6$ was studied by Rossini et al. [31], who pointed out analogies with counter-propagating boundary modes in the $\nu = 1/3$ Laughlin state. Third, it

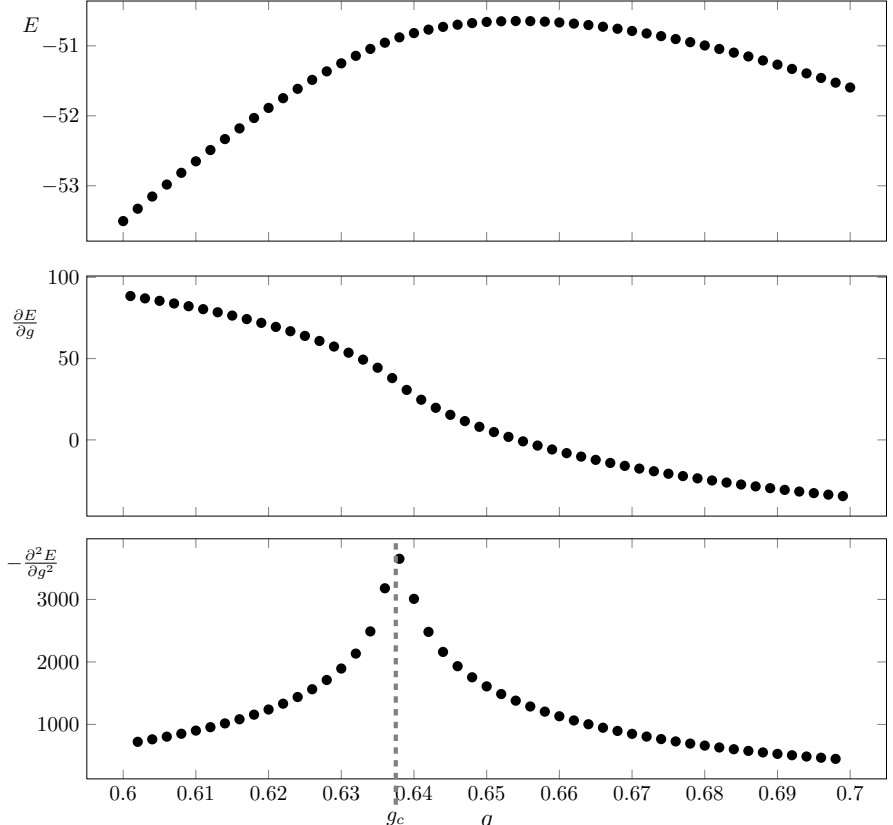

Figure 12: Ground-state energy $E(g)$ as well as its first and second order derivatives with respect to the parameter $g$ but at fixed filling $n = 0.3$. We observe a divergence in $\frac{\partial^2 E}{\partial g^2}$ at $g_c \simeq 0.64$ indicating the existence of a second-order phase transition between the L and R phase.

would be of general interest to establish possible experimental realisations of FPFs, for example based on structures combining quantum Hall systems and superconductors, quantum Hall bilayers, or two-dimensional topological insulators [52].

# Acknowledgements

We would like to thank Eddy Ardonne, Philippe Corboz, Lars Fritz, Hans Hansson, Fabian Hassler, Christoph Karrasch, Hosho Katsura, Benedikt Schoenauer and particularly Floris Elzinga for very useful discussions. IM would like to thank the Institute for Theoretical Physics at Utrecht University for hospitality during his visit where parts of this project were done.

**Funding information** IM was sponsored, in part, by the Swedish Research Council, under Grant No. 2015-05043. The work was supported by travel grants from Fonden för främjande av fysik forskning and the D-ITP. This work is part of the D-ITP consortium, a program of the Netherlands Organisation for Scientific Research (NWO) that is funded by the Dutch Ministry of Education, Culture and Science (OCW).

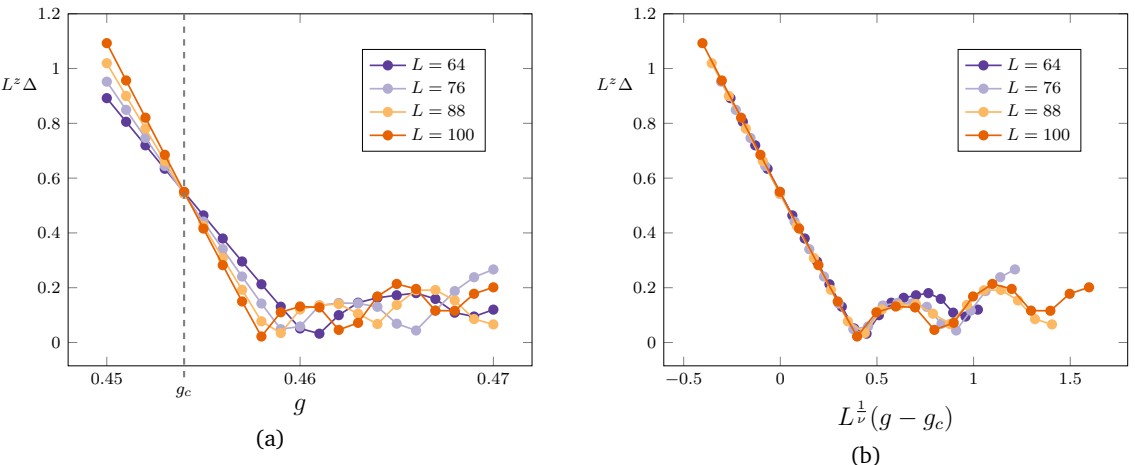

Figure 13: (a) The energy gap at $n = 1$ scaled with the system size, $L\Delta(L, g)$, indicates a phase transition at $g_c \simeq 0.45$. (b) The scaling collapse of data using the critical exponents $z = \nu = 1$.

## A  Proof of "particle-hole" symmetry

In this section we prove that it is sufficient to study the model for $0 < n \leq 1$. First of all note that the number operator for FPFs,

$$N_j = \mathbf{1} \otimes \cdots \otimes \underbrace{N}_{j} \otimes \cdots \otimes \mathbf{1} \,, \qquad N = \begin{pmatrix} 0 & 0 & 0 \\ 0 & 1 & 0 \\ 0 & 0 & 2 \end{pmatrix}, \tag{43}$$

can be rewritten in terms of the bosonic operators (20) as

$$N_j = F_j^{\dagger 2} F_j^2 + F_j^\dagger F_j = B_j^{\dagger 2} B_j^2 + B_j^\dagger B_j. \tag{44}$$

We now perform the transformation

$$U_j \to U_j^\dagger, \qquad B_j \to B_j^\dagger, \tag{45}$$

which preserves the bosonic algebra (21). From Equation (18) one can see that this transformation corresponds to $F_j \to F_j^\dagger$. Applying it to $N_j$ we get for the particle density

$$N_j \to B_j^2 B_j^{\dagger 2} + B_j B_j^\dagger = \mathbf{2} - N_j \quad \Rightarrow \quad n = \frac{1}{L} \sum_{j=1}^L N_j \to 2 - n. \tag{46}$$

The action of (45) on the Hamiltonian (22) is given by

$$H(g) = -t(1-g) \sum_{j=1}^{L-1} \left( B_j^\dagger U_j B_{j+1} + U_j^\dagger B_j B_{j+1}^\dagger \right) - tg \sum_{j=1}^{L-1} \left( B_j^{\dagger 2} B_{j+1}^2 + B_j^2 B_{j+1}^{\dagger 2} \right) \tag{47}$$

$$\to -t(1-g) \sum_{j=1}^{L-1} \left( B_j U_j^\dagger B_{j+1}^\dagger + U_j B_j^\dagger B_{j+1} \right) - tg \sum_{j=1}^{L-1} \left( B_j^2 B_{j+1}^{\dagger 2} + B_j^{\dagger 2} B_{j+1}^2 \right) \tag{48}$$

$$= -t(1-g) \sum_{j=1}^{L-1} \left( \omega B_j^\dagger U_j B_{j+1} + \bar\omega U_j^\dagger B_j B_{j+1}^\dagger \right) - tg \sum_{j=1}^{L-1} \left( B_j^{\dagger 2} B_{j+1}^2 + B_j^2 B_{j+1}^{\dagger 2} \right). \tag{49}$$

We recall that we can choose other representations for the matrix $U$ in Equation (19) as long as it satisfies the requirements $U^3 = \mathbb{1}$ and $U^2 = U^\dagger$. Thus we can redefine $U_j$ as $\tilde{U}_j = \omega U_j$, which still satisfies the algebra (21) with the $B_j$'s. Therefore Equation (49) can be rewritten in terms of $\tilde{U}_j$ and then retrieves its original form (47).

Note that although the model (15) can be defined for any $p \geq 3$, its bosonic representation (22) was written specifically for the case of $p = 3$. Hence our proof is also restricted to this case.

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
