# Peer review of "Phase diagram of the $\mathbb{Z}_3$-Fock parafermion chain with pair hopping"

_SciPost Physics Core, doi:SciPost Phys. Core 3, 011 (2020)_

## Round 1 · Referee Report · Anonymous (Referee 1) · 2020-4-9

Report

The authors investigate a chain of $Z_3$ Fock parafermions with both single particle and pair hopping terms. They use a mixture of analytical and numerical techniques to determine a phase diagram consisting of two gapless regions described by conformal field theories with central charge $c=1$, a gapless region with $c=2$ and a Mott-insulator gapped phase. Their analysis is careful and diligent, clearly indicating the differences between the phases and appears very useful in the understanding of parafermionic phases of matter, building nicely on work already done in the field. I recommend that this work is suitable for publication in SciPost, although a few minor changes could be made.

Requested changes

1) Equation (8) appears to be slightly wrong and inconsistent with Ref. [29] in the text [E. Cobanera and G. Ortiz, Fock parafermions and self-dual representations of the braid group, Phys. Rev. A 89, 012328 (2014), doi:10.1103/PhysRevA.89.012328, ibid. 91, 059901(E) (2015)]. I believe the power of $\omega$ should be changed from $m(m+1)/2$ to $m(m+3)/2$ for the two to agree and for consistency with Equation (18).

2) The correlators plotted in Figures (3) and (5) clearly show the algebraic decay described in the text but also seem to have oscillatory behaviour, particularly in (5). These oscillations seem to be quite strong in some cases and so some insight into the physical reason for these, and possibly even a quantitative understanding if possible, would be a useful addition.

3) While the phases themselves are very clearly explained, the phase transitions seem a little more uncertain. In particular, the authors show that the transition between the L and R phases appears to be second order, but a little more detail would be helpful. Does this transition line appear to be explained by a further conformal field theory, from which there is a flow to L and R in the two directions? By investigating the behaviour of the dynamical critical exponent and entanglement entropy for different lattice sizes as the transition is approached, this may be possible to determine. Alternatively, it may just be too difficult numerically.

  • validity: high
  • significance: good
  • originality: good
  • clarity: high
  • formatting: excellent
  • grammar: excellent

Author:  Dirk Schuricht  on 2020-05-13  [id 824]

(in reply to Report 1 on 2020-04-09)
Category:
answer to question

We thank the referee for his/her positive remarks and suggestions to improve our presentation. We have adapted the text accordingly, see the resubmitted version v2. Below let us comment on the specific points raised by the referee: 1. We thank the referee for his/her careful reading of the manuscript. Indeed the referee is correct that the exponent of $\omega$ in (8) was incorrect. Since our numerical simulations were performed starting from (18), all of our results remain correct. 2. We have added Figure 5(b) to make the oscillations more visible and also added discussions thereof at several places Secs. 5.1 and 5.2. The wave length of the oscillations as well as their power-law decay depend non-trivially on the filling fraction n and the parameter g. Thus it was not possible to obtain a qualitative understanding using the Luttinger-liquid description. 3. We fully agree with the referee that it would be very interesting to analyse the nature of the phase transitions in more detail. We have already performed numerical calculations to address this, see Figures 9, 12 and 13. Unfortunately the numerical simulations close to the phase transitions become quite delicate and demanding, thus we were not able to obtain more information on the phase transitions. We decided to focus on the characterisation of the phases and leave the further analysis of the transitions for the future, as we now explicitly mention in Section 5.5.1 and the outlook.

---

## Round 1 · Referee Report · Anonymous (Referee 2) · 2020-9-8

Strengths

  • The paper is written in a very pedagogical way.
  • The topic is very timely and the analysis of the phase diagram is useful.

Weaknesses

  • The phase diagram is incomplete even for the simple model considered by the authors. Complex phases give rise to different phases and should be discussed.
  • The analysis is rather straightforward and in my opinion does not meet the impact criteria for SciPost Physics.

Report

The authors study the phase diagram of the Z3 parafermionic chain. Using the recently developed concept of Fock parafermions and numerical approaches based on existing DMRG packages (ALPS and TenPy), they investigate the entanglement entropy and the finite-size scaling of the spectral gap of the model. They use this to identify four different phases as a function of the filling fraction and the relative strength of single-particle and pair hopping.

In my opinion, the paper is very nicely written and contains a useful discussion of the phase diagram. However, I do not think that this paper meets the impact criteria of SciPost Physics. In particular, it seems to me that the (mostly numerical) analysis is rather straightforward and does not rest on any novel techniques or approaches. Moreover, the model studied by the authors is rather a special case: it is well known that the complex phases of the hopping amplitudes have an important influence of the phase diagram.

The paper should ultimately be published, but in my opinion, it represents rather incremental progress in the field of parafermionic chains, so a more specialized journal might be more suitable.
  • validity: high
  • significance: ok
  • originality: low
  • clarity: high
  • formatting: excellent
  • grammar: excellent

Author:  Dirk Schuricht  on 2020-09-16  [id 973]

(in reply to Report 2 on 2020-09-08)

We thank the referee for his/her positive remarks. We agree that our analysis used well-known techniques and approaches, however, our motivation was the study of the Fock parafermion model (15), to which end these methods proved suitable. Also we agree with the referee that the study of the effect of complex phases constitutes an interesting and important extension of our analysis. However, we believe that including such an analysis would go well beyond the aim of our work, which should be seen as an initial step in the study of Fock parafermion models. We have thus extended the discussion in the outlook to clarify the relevance of this extension for future studies.

---

## Round 3 · Referee Report · Anonymous (Referee 2) · 2020-10-27

Report
It is regrettable that the authors did not include a discussion of the complex phases in their numerical analysis. Technically, this seems like a very simple extension, which would have made the paper more complete. But I respect the authors' choice not to do it and to publish this separately.
In my opinion, this paper can be accepted for publication.
In my opinion, this paper can be accepted for publication.

---

## Round 3 · List of Changes

Summary of changes in reply to both referees:
-added Reference 24
-rewording of and around (2), (3) and (4) to clarify the algebra
-added a remark after (6)
-corrected Eq. (8) as noted by the referee
-added a remark after (14) on the relation to spinful fermions in the case p=4 as well as Reference 33
-clarified the relation between (1) and (15) in a new paragraph after (15)
-added point R_2’ in FIg. 1
-specified the basis used to define (19) and (20)
-added discussion on the appearance of oscillations in Figure 3, in (33) and as well as after (34)
-added Figure 5(b), adapted the caption and added a discussion after (39)
-changed notation in (42) and Fig. 11 for consistency
-mention the study of the phase transitions as open point in Section 5.5.1 and the outlook
-corrected caption of Figure 13
-extended the discussion in the last paragraph of the outlook and added references [50,51]
-updated the acknowledgement
-corrected a few typos
-added Reference 24
-rewording of and around (2), (3) and (4) to clarify the algebra
-added a remark after (6)
-corrected Eq. (8) as noted by the referee
-added a remark after (14) on the relation to spinful fermions in the case p=4 as well as Reference 33
-clarified the relation between (1) and (15) in a new paragraph after (15)
-added point R_2’ in FIg. 1
-specified the basis used to define (19) and (20)
-added discussion on the appearance of oscillations in Figure 3, in (33) and as well as after (34)
-added Figure 5(b), adapted the caption and added a discussion after (39)
-changed notation in (42) and Fig. 11 for consistency
-mention the study of the phase transitions as open point in Section 5.5.1 and the outlook
-corrected caption of Figure 13
-extended the discussion in the last paragraph of the outlook and added references [50,51]
-updated the acknowledgement
-corrected a few typos

---

## Editorial Decision

published